# Supplementation with a New Standardized Extract of Green and Black Tea Exerts Antiadipogenic Effects and Prevents Insulin Resistance in Mice with Metabolic Syndrome

**DOI:** 10.3390/ijms24108521

**Published:** 2023-05-10

**Authors:** Mario De la Fuente-Muñoz, María De la Fuente-Fernández, Marta Román-Carmena, Sara Amor, María C. Iglesias-de la Cruz, Guillermo García-Laínez, Silvia Llopis, Patricia Martorell, David Verdú, Eva Serna, Ángel L. García-Villalón, Sonia I. Guilera, Antonio M. Inarejos-García, Miriam Granado

**Affiliations:** 1Departamento de Fisiología, Facultad de Medicina, Universidad Autónoma de Madrid, 28029 Madrid, Spain; 2Nutrition Archer Daniels Midland (ADM) Health & Wellness, Biopolis S. L. Parc Cientific, Universitat de València, 46980 Paterna, Spain; 3Departamento de Fisiología, Facultad de Medicina, Universidad de Valencia, 46010 Valencia, Spain; 4R&D Department of Functional Extracts, ADM® Valencia, 46740 Carcaixent, Spain; 5Centro de Investigación Biomédica en Red Fisiopatología de la Obesidad y Nutrición, Instituto de Salud Carlos III, 28029 Madrid, Spain

**Keywords:** obesity, metabolic syndrome, insulin resistance, green tea, black tea, extract, antioxidant

## Abstract

Insulin resistance is one of the main characteristics of metabolic syndrome (MetS) and the main cause of the development of type II diabetes. The high prevalence of this syndrome in recent decades has made it necessary to search for preventive and therapeutic agents, ideally of natural origin, with fewer side effects than conventional pharmacological treatments. Tea is widely known for its medicinal properties, including beneficial effects on weight management and insulin resistance. The aim of this study was to analyze whether a standardized extract of green and black tea (ADM^®^ Complex Tea Extract (CTE)) prevents the development of insulin resistance in mice with MetS. For this purpose, C57BL6/J mice were fed for 20 weeks with a standard diet (Chow), a diet with 56% kcal from fat and sugar (HFHS) or an HFHS diet supplemented with 1.6% CTE. CTE supplementation reduced body weight gain, adiposity and circulating leptin levels. Likewise, CTE also exerted lipolytic and antiadipogenic effects in 3T3-L1 adipocyte cultures and in the C. elegans model. Regarding insulin resistance, CTE supplementation significantly increased plasma adiponectin concentrations and reduced the circulating levels of insulin and the HOMA-IR. Incubation of liver, gastrocnemius muscle and retroperitoneal adipose tissue explants with insulin increased the pAkt/Akt ratio in mice fed with Chow and HFHS + CTE but not in those fed only with HFHS. The greater activation of the PI3K/Akt pathway in response to insulin in mice supplemented with CTE was associated with a decrease in the expression of the proinflammatory markers *Mcp-1*, *IL-6*, *IL-1β* or *Tnf-α* and with an overexpression of the antioxidant enzymes *Sod-1*, *Gpx-3*, *Ho-1* and *Gsr* in these tissues. Moreover, in skeletal muscle, mice treated with CTE showed increased mRNA levels of the aryl hydrocarbon receptor (*Ahr*), *Arnt* and *Nrf2*, suggesting that the CTE’s insulin-sensitizing effects could be the result of the activation of this pathway. In conclusion, supplementation with the standardized extract of green and black tea CTE reduces body weight gain, exerts lipolytic and antiadipogenic effects and reduces insulin resistance in mice with MetS through its anti-inflammatory and antioxidant effects.

## 1. Introduction

Obesity is considered the pandemic of the 21st century. According to data from the World Health Organization (WHO), in 2016 more than 1.9 billion adults were overweight and, among these, over 650 million were obese [1]. This pandemic affects not only developed countries but also developing ones, where the economic health burden has dramatically risen due to the increased incidence of cardiometabolic alterations associated with this condition [2].

Obesity is one of the main risk factors for the development of a more complex condition named metabolic syndrome (MetS), which includes abdominal obesity, insulin resistance, impaired fasting glucose, hypertension, hypertriglyceridemia and low HDL cholesterol levels [3]. The global prevalence of MetS is estimated to be about one quarter of the world’s population, in other words, over a billion people in the world are now affected with MetS [4].

One of the main characteristics of MetS is insulin resistance, a condition that is present in more than 80% of the patients and that is associated with impaired insulin sensitivity in insulin-dependent tissues such as the liver, adipose tissue and the skeletal muscle [5]. This is associated with a decreased activation of the PI3K/Akt pathway in these tissues, resulting in reduced translocation of GLUT-4 transporters to the cell membrane and/or reduced glucose storage in the form of glycogen [6].

Insulin resistance is reported to be one of the major contributors to the development of metabolic alterations such as dyslipidemia, visceral adiposity, hyperuricemia and hyperglycemia that lead, in the long term, to the development of metabolic diseases such as type II diabetes [7]. Likewise, decreased insulin sensitivity produces cardiovascular alterations, such as endothelial dysfunction and a prothrombic state, that increase the risk of suffering cardiovascular diseases including hypertension, arrhythmias, stroke, or coronary heart disease [8].

Although the physiopathological mechanisms involved in the development of insulin resistance vary among tissues, it is well known that lipotoxicity, a phenomenon caused by an excessive accumulation of lipid intermediates, such as triglycerides, diacylglycerols (DAGs), or ceramides, is a common feature among all of them [9,10]. These lipid intermediates induce several alterations, such as inflammation, increased reactive oxygen species (ROS) production and mitochondrial dysfunction, which lead, among other consequences, to an impaired fatty acid and glucose oxidation and increased fatty acid release [5].

The complexity and the large number of mediators implicated makes it difficult to find a single and effective treatment to alleviate the development of cardiometabolic alterations associated with MetS. For this reason, therapeutic approaches for this syndrome are based on the combination of several drugs that include, among others, antihypertensives, insulin sensitizers and antilipidemic and antithrombotic agents. These treatments are of high cost and are not exempt from interactions and side effects. For this reason, in recent decades a great effort has been made in the search for new strategies, ideally of natural origin, that are effective in the treatment and prevention of MetS. Since oxidative stress is one of the most relevant pathophysiological mechanisms in the development of MetS-associated metabolic disorders, antioxidant substances, such as vitamins E and C, have been proposed as possible therapeutic candidates for its treatment [11,12,13]. Likewise, MetS is usually associated with increased levels of inflammatory mediators such as C reactive protein (PCR), interleukin-6 (IL-6), or tumoral necrosis factor alpha (TNF-α) [14], both in the plasma and at the tissue level, so molecules with anti-inflammatory effects may also be potentially useful.

Tea consumption is reported to exert many beneficial health effects [15], and specifically for the treatment and/or prevention of the metabolic alterations associated with MetS, both in experimental animals [16,17,18,19] and in humans [20,21]. Particularly, consumption of both green and black tea reduces body weight and exerts lipolytic and antiadipogenic effects as well as insulin-sensitizing activity [22,23]. In green tea, these effects have been attributed to the presence of monomeric catechins such as epicatechin, epigallocatechin, epicatechin gallate and epigallocatechin gallate [24], whereas in black tea these effects are due to thearubigins and mostly to theaflavins [25]. However, the lack of standardized formulations makes it difficult to compare the effects among different brands and types of tea, including infusions and tea extracts, as each of them contains different types and concentrations of bioactive substances.

In a previous study from our group, we reported that supplementation with the standardized extract of green and black tea, ADM^®^ Complex Tea Extract (CTE), prevents endothelial dysfunction and the development of hypertension in mice with MetS through its anti-inflammatory and antioxidant properties [26]. However, the effects of supplementation with this standardized tea extract on metabolism remain to be determined. For this reason, the aim of this study was to evaluate if supplementation with the standardized extract of green and black tea CTE prevents MetS-induced insulin resistance and, if so, to explore the possible mechanisms involved. For this purpose, insulin sensitivity as well as the expression of several inflammatory and oxidative stress markers were assessed in the liver, gastrocnemius and retroperitoneal adipose tissue from lean control mice and from obese mice with MetS. In addition, since the aryl hydrocarbon receptor (AhR) pathway is reported to be involved in the antioxidant and anti-inflammatory effects of some of the bioactive substances present in CTE, such as gallic acid, xanthines and flavonols in MetS [27], the activation of this pathway was studied as a possible mechanism.

## 2. Results

### 2.1. Individual Composition of CTE (Liquid Chromatography)

The individual composition of the phenolic components of the standardized ADM^®^ Complex Tea Extract analyzed by HPLC-PAD is depicted in Table 1. The major monomeric flavan-3-ols were EGCg (6.0%, dry basis) and Epicatechin-3-gallate (2.5%, dry basis). Regarding oligomeric flavan-3-ols, theaflavin was the most concentrated, and the total content of theaflavins was about 0.1% (dry basis). Within methylxanthines, caffeine comprised 96% of all xanthines (6.5%, dry basis). The content of total bioactive components (flavan-3-ols, methyl xanthines and gallic acid) was about 20.4%. The certificate of analysis of the manufacturer considers Flavan-3-ols (monomeric and theaflavins), together with total xanthines (caffeine, theobromine and theophylline), as the main quality markers of Complex Tea Extract.

### 2.2. Volatile Compounds Detected by Gas Chromatography

The headspace SPME-GC-FID/MS analysis allowed for the identification of characteristic volatile components from CTE. The quantification was performed according to their relative response proportion. CTE showed Group I and Group II volatile compounds in CTE (Appendix A). The major components were linalool (14.8%), linalool oxides (2.6%) and benzaldehyde (2.2%).

Thus, Complex Tea Extract shows both complete phenolic and volatile profiles from green and black teas, with the main components from these teas shown in Table 1 for polar phenolics and Appendix A for volatile compounds.

### 2.3. In Vitro Adipogenesis

3T3-L1 preadipocyte differentiation into adipocytes induced an increase in lipid content measured by Oil Red O staining (*p* < 0.001; Figure 1A). Epigallocatechin gallate (100 µM) and CTE at 0.4 and 0.5 mg/mL reduced the differentiation-associated increase in lipid content (*p* < 0.001 for all), but not CTE, at 0.1 and 0.25 mg/mL. No statistically significant differences were found in the lipid content of differentiated adipocytes between 0.4 and 0.5 mg/mL CTE and the positive control (epigallocatechin gallate)-treated cells.

### 2.4. Fat and Triglyceride Content in C. elegans

The consumption of any tested dose of CTE and 6 µg/mL of Orlistat reduced fat content in *C. elegans* compared to control NGM (*p* < 0.001 for all; Figure 1B). Nevertheless, Orlistat induced a higher reduction in fat content than all CTE dosages (*p* < 0.001 for all).

As shown at Figure 1C, young adult nematodes fed with 6 µg/mL of Orlistat and 2 mg/mL of CTE showed reduced triglyceride content compared to control NGM worms (*p* < 0.01 and *p* < 0.05, respectively). Orlistat induced a higher reduction in triglyceride amount than CTE treatment (*p* < 0.05).

### 2.5. Body Weight Gain, Food and Caloric Intake in Mice with MetS

The consumption of the HFHS diet for 20 weeks induced a significant increase in body weight gain (*p* < 0.001; Figure 2A,B), and this increase was significantly lower in obese mice supplemented with CTE (*p* < 0.05).

In mice fed with the HFHS diet, accumulated food intake was decreased (Figure 2B) and accumulated caloric intake was increased (Figure 2C) compared to lean animals regardless of the supplementation with CTE (*p* < 0.001 for both).

### 2.6. Organ Weights

Compared to controls, mice fed the HFHS diet showed increased weights of liver (*p* < 0.001; Table 2), spleen (*p* < 0.01), kidneys (*p* < 0.001) and adrenal glands (*p* < 0.05). Likewise, the weight of soleus muscle (*p* < 0.05) and the weights of all adipose tissue (AT) depots, both white (visceral epidydimal, visceral retroperitoneal and subcutaneous lumbar) and brown (interscapular and periaortic), were significantly higher in mice fed the HFHS diet compared to mice fed the standard chow (*p* < 0.001 for all).

Supplementation with CTE did not affect the weights of liver, kidneys, adrenal glands, epidydimal AT, periaortic AT and soleus and gastrocnemius muscles, but it significantly reduced the weights of spleen (*p* < 0.05), retroperitoneal AT (*p* < 0.001), lumbar AT (*p* < 0.05) and interscapular AT (*p* < 0.001).

### 2.7. Plasma Measurements

As shown in Table 3, obese mice fed with the HFHS diet for 20 weeks showed increased circulating levels of insulin, adiponectin and leptin as well as increased glycaemia and HOMA-IR compared to lean mice (*p* < 0.001 for all). Supplementation with CTE did not prevent the obesity-induced increase in glycaemia but it significantly reduced the plasma concentrations of leptin (*p* < 0.01), adiponectin (*p* < 0.05) and insulin (*p* < 0.05), as well as the HOMA-IR (*p* < 0.05).

### 2.8. Insulin Sensitivity and Gene Expression of Inflammatory and Oxidative Stress Markers in the Liver

Hepatic insulin sensitivity was measured by the ratio of phosphorylated Akt and total Akt as an index of activation of the PI3K/Akt pathway in liver explants incubated in the presence/absence of insulin for 15 min. In addition, the gene expression of both *Ins-r* and the enzyme *Gys-1* was also assessed in the hepatic tissue of mice from the different experimental groups. The results are represented in Figure 3A,B, respectively.

Incubation of liver explants with insulin induced a significant increase in the pAkt/Akt ratio in control mice (*p* < 0.001) and in obese mice supplemented with CTE (*p* < 0.001), but not in untreated obese mice. In addition, there was a downregulation of the gene expression of both *Ins-r* (*p* < 0.01) and *Gys-1* (*p* < 0.05) in the hepatic tissue of HFHS mice compared to controls that was totally prevented by CTE supplementation (*p* < 0.01 and *p* < 0.05, respectively).

Decreased insulin sensitivity in the liver of untreated obese mice was associated with an upregulation of the gene expression of the inflammatory markers *Mcp-1* (*p* < 0.01; Figure 4A), *IL-1β* (*p* < 0.01; Figure 4A), *IL-6* (*p* < 0.01; Figure 4A), *IL-10* (*p* < 0.01; Figure 4A) and *Tnf-α* (*p* < 0.01; Figure 4A), and with an upregulation of the mRNA levels of the prooxidant enzyme *Nox-1* (*p* < 0.01; Figure 4B). On the contrary, the gene expression of the antioxidant enzymes *Nox-4*, *Sod-1* and *Gsr* was significantly downregulated in the liver of HFHS mice (*p* < 0.05 for all; Figure 4B).

Supplementation with CTE did not prevent the obesity-induced changes in the mRNA levels of *IL-10* and *Gsr*, but it prevented the increase in *Mcp-1* (*p* < 0.01), IL-1β (*p* < 0.05), *IL-6 (p* < 0.05), *Tnf-α* (*p* < 0.05) and *Nox-1* (*p* < 0.05) as well as the obesity-induced decrease in the gene expression of *Nox-4* and *Sod-1* (*p* < 0.05 for both).

Finally, the mRNA levels of *Gpx-3* and *Ho-1* were unchanged among experimental groups.

### 2.9. Insulin Sensitivity and Gene Expression of Inflammatory and Oxidative Stress Markers in Gastrocnemius Muscle

The incubation of gastrocnemius muscle explants in the presence/absence of insulin for 15 min resulted in an increased pAkt/Akt ratio in control mice (*p* < 0.001; Figure 5A), untreated obese mice (*p* < 0.01; Figure 5A) and obese mice supplemented with CTE (*p* < 0.05; Figure 5A). However, insulin induced a higher activation of the PI3K/Akt pathway in obese mice supplemented with CTE compared to untreated obese mice (*p* < 0.05).

Mice with MetS showed a significant decrease in the gene expression of *Ins-r* (*p* < 0.05; Figure 5B) and *Gys-1* (*p* < 0.01; Figure 5B) in gastrocnemius muscle compared to controls, and these effects were totally prevented by supplementation with CTE (*p* < 0.05 for both; Figure 5B).

In gastrocnemius muscle, no changes were found among experimental groups in the gene expression of *IL-6* and *Nox-1*. However, obesity was associated with an overexpression of the inflammatory markers *Mcp-1* (*p* < 0.01: Figure 6A), *IL-1β* (*p* < 0.05: Figure 6A), *IL-10* (*p* < 0.05: Figure 6A) and *Tnf-α* (*p* < 0.05: Figure 6A) and with a downregulation of the gene expression of the antioxidant enzymes *Nox-4*, *Sod-1*, *Gpx-3* and *Ho-1* (*p* < 0.05 for all: Figure 6B). All these changes were prevented by supplementation with CTE.

### 2.10. Adipocyte Size and Gene Expression of Enzymes Related to Lipidic Metabolism in Retroperitoneal Adipose Tissue

Figure 7A shows the mean area of adipocytes from retroperitoneal adipose tissue of mice from the different experimental groups.

The mean area of adipocytes was significantly higher in the adipose tissue of untreated obese mice compared to controls (*p* < 0.001), and this effect was attenuated by supplementation with CTE (*p* < 0.01).

The mRNA levels of different markers and enzymes related to lipid metabolism are shown in Figure 7B.

Obesity was associated with a significant reduction in the gene expression of *Fasn* (*p* < 0.05), *Lpl* (*p* < 0.01), *Hsl* (*p* < 0.001), *Ppar-γ* (*p* < 0.001), *Pgc-1α* (*p* < 0.001), *Ucp-1* (*p* < 0.05), *β3-Adr* (*p* < 0.001) and *Ob-r* (*p* < 0.05). Supplementation with CTE did not increase the mRNA levels of *Pgc-1α* but it attenuated the obesity-induced downregulation of *Fasn* (*p* < 0.01), *Lpl* (*p* < 0.05), *Hsl* (*p* < 0.01), *Ppar-γ* (*p* < 0.05), *Ucp-1* (*p* < 0.05), *β3-Adr* (*p* < 0.05) and *Ob-r* (*p* < 0.05).

### 2.11. Insulin Sensitivity and Gene Expression of Inflammatory and Oxidative Stress Markers in Retroperitoneal Adipose Tissue

As shown in Figure 8A, the incubation of retroperitoneal AT explants with insulin induced a significant increase in the pAkt/Akt ratio in control mice (*p* < 0.01) and in obese mice supplemented with CTE (*p* < 0.05) but not in untreated obese mice. In addition, obesity was associated with a downregulation of the gene expression of *Ins-r* (*p* < 0.001; Figure 8B) that was partially prevented by CTE supplementation (*p* < 0.01; Figure 8B).

Obesity was associated with an upregulation of the gene expression of the inflammatory markers *Mcp-1* (*p* < 0.01; Figure 9A), *IL-6* (*p* < 0.01; Figure 9A), *IL-10* (*p* < 0.01; Figure 9A) and *Tnf-α* (*p* < 0.001; Figure 9A), and with an upregulation of the mRNA levels of *Nox-1* (*p* < 0.05; Figure 9B) and *Ho-1* (*p* < 0.001; Figure 9B). On the contrary, the gene expression of the antioxidant enzymes *Sod-1*, *Gpx-3* and *Gsr* was significantly downregulated in the liver of HFHS mice (*p* < 0.001 for all; Figure 9B).

Supplementation with CTE did not prevent the obesity-induced changes in the mRNA levels of *IL-10* and *Nox-1* but it prevented the increase in *Mcp-1* (*p* < 0.05), *IL-1β* (*p* < 0.05), *IL-6* (*p* < 0.05), *Tnf-α* (*p* < 0.05), *Nox-1* (*p* < 0.05) and *Ho-1* (*p* < 0.05) as well as the obesity-induced decrease in the gene expression of *Sod-1* (*p* < 0.001), *Gpx-3* (*p* < 0.05) and *Gsr* (*p* < 0.05).

### 2.12. mRNA Levels of Ahr, Arnt and Nrf2 in the Liver, Gastrocnemius Muscle and Retroperitoneal Adipose Tissue

The gene expressions of *Ahr, Arnt* and *Nrf2* in the liver, gastrocnemius muscle and retroperitoneal adipose tissue are represented in Figure 10A–C, respectively.

Obesity induced a significant decrease in the mRNA levels of *Ahr* (*p* < 0.001; Figure 10C) and *Arnt* (*p* < 0.05; Figure 10C) in retroperitoneal AT but not in the liver and the gastrocnemius muscle. In addition, the gene expression of *Nrf2* was downregulated both in the gastrocnemius muscle (*p* < 0.05: Figure 10B) and in the retroperitoneal AT (*p* < 0.05; Figure 10C) of obese mice compared to controls.

CTE supplementation prevented the obesity-induced reduction in the mRNA levels of *Ahr* in retroperitoneal AT (*p* < 0.05) and significantly increased the gene expression of *Arnt* in gastrocnemius muscle (*p* < 0.05).

Moreover, in order to verify whether the beneficial effects of CTE supplementation observed in the HFHS model could be related to the activation of the *AhR* pathway, a study of correlations between the different genes was performed. The correlation study shows a positive and significant correlation between *Ahr* and *Arnt* (R = 0.90; *p* value < 0.05), but not for *Ahr* and *Nrf2* (R = 0.14; *p* value > 0.05), in adipose tissue. Furthermore, a positive correlation was found between the AhR pathway genes *Arnt* and *Nrf2* (Appendix A; *p* < 0.05 for both) as well as for the antioxidant genes *Sod-1, Gpx-3* and *Gsr* in gastrocnemius muscle (Appendix A; *p* < 0.001 for all). On the contrary, there was a negative correlation between Mcp-1 and Ahr mRNAs (Appendix A; *p* < 0.05).

## 3. Discussion

In this paper, the beneficial effects of the standardized green and black tea extract ADM^®^ Complex Tea Extract (CTE) on the metabolic alterations associated with MetS are reported.

The results of the in vivo study show that supplementation with CTE reduces both body weight gain and adiposity in mice fed a high fat/high sucrose diet. These results are in agreement with previous studies, in which supplementation with green or black tea extracts showed a positive effect of decreasing body fat and reducing body weight both in experimental animals and in humans [28,29,30,31,32,33]. However, there are also several studies, especially in humans, that have not found significant effects on body composition after supplementation with different tea extracts [34,35,36,37]. These discrepancies among studies may be due to differences in the type of tea, intake levels, time of treatment, species or even ethnicity [38]. In addition, discrepancies are also most likely due to differences in the composition of formulations, since the amount of bioactive compounds varies substantially between different tea extracts and infusions. For this reason, the production of tea formulations with a standardized composition of bioactive substances is highly recommended.

In our study, CTE exerted antiadipogenic effects in 3T3-L1 adipocytes where it significantly reduced the differentiation of preadipocytes into mature adipocytes. Moreover, CTE supplementation also reduced adiposity in vivo, both in the *C. elegans* model and in the murine model of MetS. The effects of CTE supplementation on adiposity in mice with MetS were associated with a significant decrease in the size of adipocytes and with an upregulation of the mRNA levels of lipolytic markers such as *Hsl*, adrenergic β3 receptor or leptin receptor in adipose tissue. Moreover, supplementation with CTE prevented the obesity-induced decrease in the gene expression of *Ucp-1*, possibly indicating an increase in thermogenesis. The positive effects of both green and black tea in reducing adiposity have already been demonstrated both in vivo and in vitro [39] and seem to be mediated mainly by catechins [40], particularly (−)-catechin 3-gallate and (−)-epigallocatechin 3-gallate, which are reported to inhibit the insulin-induced glucose uptake by 3T3-L1 adipocytes in a dose- and time-dependent manner [41]. Moreover, catechins not only affect glucose uptake by adipocytes, but also lipid storage by inducing lipolysis through HSL activation [42] and promoting a calorigenic action through UCP-1 upregulation [43]. In addition to catechins, another possible contributor to the antiadipogenic and lipolytic effects of CTE is caffeine, which also exerts a marked effect on lipid metabolism both alone [44,45] and in combination with catechins [43,46].

In addition to the effects on body composition, probably the most important finding of this study is that supplementation with CTE in mice fed an HFHS diet attenuates the development of insulin resistance. Indeed, despite changes in glycaemia, CTE-treated animals showed decreased insulin concentrations and HOMA-IR compared to obese untreated mice. Moreover, CTE supplementation prevented the obesity-induced reduction in adiponectin circulating levels, an effect that has been also reported in different clinical trials [47]. These results are in agreement with previous studies that have reported insulin-sensitizing effects of green tea both in experimental animals [48,49,50] and in obese [51,52] and diabetic [53] patients. However, there are also studies, mainly clinical trials, that did not find significant insulin-sensitizing effects after tea supplementation [54,55]. These negative results may be related to methodological aspects related to the number of participants, the type of extracts employed and/or the dosages administered. Indeed, the dosages employed in animal models are usually significantly higher than those administered to humans [56].

Since insulin resistance is the result of decreased insulin sensitivity in metabolic tissues such as skeletal muscle, adipose tissue and liver, we studied the activation of the PI3K/Akt pathway in these tissues in response to insulin. Our results show that obesity is associated with decreased insulin sensitivity in the three tissues since the incubation of explants with insulin did not activate the PI3K/Akt pathway. On the contrary, explants from lean animals and from obese animals supplemented with CTE showed an increased pAkt/Akt ratio in response to insulin. In addition, the increased pAkt/Akt ratio was also associated with increased mRNA levels of both insulin receptors in all tissues and with increased gene expression of glycogen synthase 1 in the liver and in gastrocnemius muscle. These results clearly show a preserved insulin sensitivity in obese mice supplemented with CTE.

To further investigate the mechanism by which CTE prevents the development of insulin resistance in mice fed an HFHS diet, and considering that decreased insulin sensitivity is highly related to increased inflammation and ROS production in metabolic tissues, we determined the mRNA levels of different proinflammatory and oxidative stress markers in the liver, gastrocnemius muscle and adipose tissue. Our results indicate that CTE supplementation prevents the obesity-induced upregulation of several proinflammatory markers such as *Mcp-1*, *IL-6*, *IL-1β* or *Tnf-α.* These results agree with our previous study, in which supplementation with CTE decreased the gene expression of proinflammatory cytokines in arterial tissue from mice with MetS [26]. Both green and black tea contain bioactive substances with anti-inflammatory properties [57]. The anti-inflammatory effects of green tea are widely described [58] and are reported to be due mainly to epigallocatechin-3-gallate [58,59]. Likewise, there are also several studies reporting the anti-inflammatory effects of black tea both in vitro [60,61] and in vivo [62,63]. This effect is mediated, at least in part, by theaflavins, the main functional component in black tea, which have been reported to reduce the production of proinflammatory cytokines both at the central level [64] and in peripheral tissues such as the liver [65] and the skeletal muscle [66].

In addition to the beneficial effects of reducing inflammation, our results also show a positive effect of CTE in the oxidative status of the animals, which is demonstrated by a decreased expression of the pro-oxidant enzyme *Nox-1* and with an upregulation of the mRNA levels of several antioxidant enzymes, such as *Ho-1*, *Sod-1*, *Gpx-3* and *Gsr*, in the liver, adipose tissue and skeletal muscle. These results clearly demonstrate the antioxidant properties of this extract, as we have already described in the aortic tissue of mice with MetS, and it is related to its antihypertensive activity [26]. The antioxidant effects of both green and black tea have been widely reported and are also mainly due to catechins and theaflavins [67,68].

Since the beneficial effects of some of the bioactive substances present in CTE, and particularly in MetS [27], are reported to be mediated by the activation of the AhR/ARNT pathway, the activation of this pathway was explored as a possible mechanism of action. Our results show that, in adipose tissue, CTE supplementation prevented the MetS-induced decrease in the gene expression of *Ahr*. Since *Ahr* is a negative regulator of lipogenesis and adipogenesis [69,70,71], the increased gene expression of *Ahr* in CTE-supplemented mice may be related to its antiadipogenic effects. Moreover, in skeletal muscle, the positive effects of CTE preventing MetS-induced insulin resistance, reducing the gene expression of proinflammatory markers and upregulating the mRNA levels of antioxidant enzymes are associated with an overexpression of *Ahr*, *Arnt* and *Nrf2.* Indeed, as previously reported after resveratrol supplementation [72], the upregulation of *AhR* and *ARNT* genes in CTE-treated mice is associated with a positive correlation with *NRF2* and with the gene expression of antioxidant enzymes *Sod-1*, *Gpx-3* and *Gsr*. Furthermore, we observed that higher *Ahr* mRNA levels correlate with lower *Mcp-1* gene expression, demonstrating that the anti-inflammatory, antioxidant and insulin-sensitizing effects of CTE are most likely mediated by the activation of the AhR-Arnt-Nrf2 pathway. However, the role of Ahr activation in glucose homeostasis is controversial since activation of this pathway has also been reported to impair insulin secretion and to promote deleterious effects in insulin-dependent tissues such as the liver, skeletal muscle and adipose tissue [73]. In this regard, AhR deficiency is reported to enhance insulin sensitivity and reduce PPAR-α pathway activity in mice [74]. However, there is also evidence about AhR agonists protecting against obesity-related insulin resistance in obese [75] and diabetic animals [76]. Thus, it is possible that the AhR/ARNT pathway has a different role in physiological and pathological contexts and in response to different substances including nutraceuticals. Indeed, several natural products are reported to exert their biological effects through this pathway, making the AhR an interesting target for the development of new therapeutic agents to treat/prevent chronic diseases associated with increased oxidative stress and inflammation.

In conclusion, supplementation with the standardized extract of green and black tea CTE reduces body weight gain, exerts lipolytic and antiadipogenic effects and reduces insulin resistance in mice with MetS. The beneficial mechanism of CTE on the prevention of insulin resistance could be mediated, at least in part, by increasing the transcription of antioxidant genes and reducing inflammatory genes *1* through the activation of the AhR pathway.

## 4. Materials and Methods

### 4.1. Reagents and Chemicals for Chromatographic Analyses

The following standards were used for liquid chromatography analyses: xanthines (caffeine, theobromine and theophylline), monomeric flavan-3-ols ((+)-catechin, catechin gallate, (−)-Epicatechin, Epicatechin-3-gallate, epigallocatechin, epigallocatechin-3-gallate (EGCg), green tea catechin mix), theaflavins (theaflavin, theaflavins mix (tea extract from Camellia sinensis)) and gallic acid. They were purchased from Merck (Barcelona, Spain) and Phytolab (Vestenbergsgreuth, Germany) for the identification and/or quantification of characteristic bioactive components from tea (*Camellia sinensis*). Trifluoroacetic acid, acetic acid, acetonitrile, dimethyl sulfoxide and water of chromatographic quality were purchased from VWR (Barcelona, Spain).

Regarding gas chromatography, the standards used were as follows: 1-penten-3-ol, n-hexanal, n-hexanol, cisZ-3-hexenal, transE-2-hexenal, cisZ-3-hexenol, transE-2-hexenol and pentanol, and for Group II, linalool, linalool oxides, methyl salicylate, phenyl acetaldehyde, geraniol, benzyl alcohol, 2-phenylethanol, benzaldehyde, α-ionone and β-ionone were purchased from Sigma Aldrich, Steinheim, Germany. Methanol, acetic acid and furfural were obtained from VWR, Darmstadt, Germany.

### 4.2. Commercial Tea Extract

As raw materials, certified Chinese green and black tea leaves were obtained from ADM^®^ (Valencia, Spain). The industrial-scale product, ADM^®^ Complex Tea Extract (CTE), standardized to total flavan-3-ols (monomeric and theaflavins) and methylxanthines (caffeine, theobromine and theophylline), and directly obtained by water extraction of a proprietary blend of green and black tea leaves as previously described [26], was used in this study.

### 4.3. Standards Preparation

Four working calibration standards solutions (EGCg, caffeine, theaflavin and gallic acid) were employed for the quantification of the main groups of components in the tea functional powdered extracts.

### 4.4. High-Performance Liquid Chromatography (HPLC)

Flavan-3-ols, methylxanthines, theaflavins and gallic acid analyses were performed according to Lee and Ong et al. [77]. The HPLC equipment used for the analysis consisted of a Shimadzu NEXERA XR UHPLC 70 MPa coupled to a photodiode array detector SPD-M40 model (Izasa Scientific, Madrid, Spain). The chromatographic analyses were performed by an octadecyl silane column ZORBAX ECLIPSE PLUS C18 (250 mm, 4.6 mm, 5 µ) together with its corresponding precolumn (Agilent Technologies, Barcelona, Spain).

Detection was performed at 275 nm, the temperature of the oven was set at 32 °C, the work flow was maintained at 1.0 mL/min and the injection volume 2 µL. Binary gradient system used for the chromatographic separation consisted of Phase (A) 5% (*v*/*v*) acetonitrile 0.035 (*v*/*v*) trifluoroacetic acid, and Phase (B) 50% (*v*/*v*) acetonitrile 0.025% (*v*/*v*) trifluoroacetic acid. The initial conditions were set with A–B (90:10), and the gradient slightly increased to 20% at 10 min, then to 40% from 25 to 27 min. Finally, the column was again balanced to the initial gradient conditions for 3 min before the next injection.

Identification of monomeric and oligomeric flavan-3-ols (gallocatechin, epigallocatechin, catechin, epicatechin, epigallocatechin-3-gallate, gallocatechin-3-gallate, epicatechin-3-gallate and catechin-3-gallate), theaflavins or oligomeric flavan-3-ols (theaflavin, theaflavin-3-monogallate, theaflavin-3′-monogallate and theaflavin-3,3′-gallate), methylxanthines (theobromine, theophylline and caffeine) and gallic acid was performed by comparing retention time, UV-Vis and the corresponding reference standards. Quantification was carried out by external calibration curve with at least 5 different concentration points (r^2^ = 0.99), the results were expressed in percentage (%, dry basis). The sum of monomeric flavan-3-ols, methylxanthines and theaflavins were, respectively, quantified as EGCg, caffeine and theaflavin equivalents.

### 4.5. Headspace Gas Chromatography Coupled to FID and Mass Detector (SPME-GC-FID/MS)

The CTE samples dissolved in water (100 mg/mL) were placed into a 20 mL headspace vial (0.5 g of tea extract in 5 mL or 0.2 g of powdered extract in 2 mL) sealed with a silicone septum. Water blanks were analyzed at the beginning of the sequence and between samples. By exposing the SPME fiber (2 cm; divinylbenzene/carboxen/polydimethylsiloxane 50 µm/30 µm for both latter coatings; Supelco Sigma-Aldrich, Bellefonte, PA, USA) to the headspace, sampling was performed at 60 °C for 60 min. Then, the adsorbed volatile components were desorbed from the fiber at 250 °C during 60 s and transferred by splitless injection (nitrogen at 1.0 mL/min) (GC6890N + MS5973Network, Agilent, Waldbronn, Germany) on a 60 m HP-Innowax column (0.25 mm i.d., 0.25 µm polyethylenglycol film thickness, 60 m length, Agilent). The oven temperature ramp was as follows: 50 °C for 5 min, 3 °C/min up to 110 °C (0 min) and 5 °C/min up to 230 °C (31 min). The FID was set at 250 °C, the MSD transfer line at 280° C, the temperature of the ionization source at 230 °C, the quadrupole at 150 °C, the electron multiplier tube to 70 eV and the scan to m/z 40–250 (acquisition rate of 10 spectral matches). The identification was performed by the NIST05 EI Database (National Institute of Standards and Technology, Gaithersburg, MD, USA) and confirmed by authentic reference standards. The ratio of individual components was expressed as relative area percentages. Target volatiles were classified into Groups I (1-penten-3-ol, n-hexanal, n-hexanol, cis-3-hexenal, trans-2-hexenal, cis-3-hexenol, trans-2-hexenol and pentanol) and II (linalool, linalool oxides, methyl salicylate, phenylacetaldehyde, geraniol, benzyl alcohol, 2-phenylethanol, benzaldehyde, α-ionone and β-ionone). Depending on the sample, other relevant volatiles for *Camellia sinensis* (limonene, hotrienol, safranal, benzyl acetate and eugenol) were also detected.

### 4.6. In Vitro Adipocyte Differentiation

To assess the effects of CTE on the differentiation of preadipocytes to adipocytes, the in vitro model of adipogenesis was carried out using the 3T3-L1 preadipocyte cell line (ATCC CL-173TM; American Type Culture Collection; Rockville, MD, USA).

Cells in exponential growth were seeded at a density of 4 × 10^4^ cells/well in 24-well plates in complete medium (DMEM supplemented with 10% Fetal Bovine Serum (FBS), 2 mM L-Glutamine, 100 U/mL Penicillin and 100 mg/mL Streptomycin). Two days after reaching confluence, cells were induced to differentiate upon a three-day treatment using 0.5 mM 3-isobutyl-1-methylxanthine (IBMX), 10 µg/mL insulin, 1 µM dexamethasone and 100 µM ascorbic acid phosphate in complete medium. Then, cell culture medium was replaced, and cells were further cultured in complete medium containing 10 µg/mL insulin and 100 µM ascorbic acid phosphate for 3 more days. Thereafter, medium was replaced, and cells were cultured in complete medium until reaching 15 days. During the differentiation process, the cells were treated with CTE at the selected doses. CTE was added in every replacement of cell culture medium. An amount of 100 µM epigallocatechin gallate was used as a positive control.

Cell culture reagents were obtained from Thermo Fisher (Waltham, MA, USA). IBMX, insulin, dexamethasone, ascorbic acid phosphate and epigallocatechin gallate were from Merck Millipore (Burlington, MA, USA). Differentiation to adipocytes was monitored by Oil Red O (Merck Millipore; Burlington, MA, USA) staining. For this purpose, cells were washed twice with PBS and monolayers were fixed upon incubation with a 3.7% formaldehyde cold solution for 1 h. Cells were stained with 0.2% Oil Red solution in 60% (*v*/*v*) isopropanol for 30 min at room temperature. Then, cells were washed several times with water to remove unbound staining, and finally the plates were air-dried. Oil Red staining was extracted with 0.2 mL of isopropanol with gently shaking and absorbance was recorded at 492 nm.

### 4.7. Fat Reduction Assays in C. elegans

*Caenorhabditis elegans* N2 wild-type strain (Bristol) were provided by Caenorhabditis Genetic Center (CGC), University of Minnesota (Minneapolis, MN, USA). Nematodes were routinely prepared on Nematode Growth Medium (NGM) plates with *Escherichia coli* strain OP50 as a food source at 20 °C.

Fat content in *C. elegans* was measured using the Nile red (Sigma Aldrich; St Louis, MO, USA) staining method as previously described [78]. Synchronized worms were incubated in NGM plates for three days until reaching young adult stage in presence of Nile red dye in the surface at 0.05 µg/mL. ADM^®^ Tea Complex was added to the surface of NGM plates at final concentrations of 0.1, 0.5, 1 and 2 mg/mL, while Orlistat was used as a positive control at 6 µg/mL. After treatment period, nematodes were transferred to M9 buffer and the fluorescence was measured using an FP-6200 system (JASCO Analytical Instruments; Easton, MD, USA) with an excitement wavelength of 480 nm and an emission wavelength of 571 nm. Four experiments were performed per condition to analyze a total of 60 nematodes per treatment.

### 4.8. Triglyceride (TG) Quantification in C. elegans

Total TGs were quantified in age-synchronized worm nematodes cultured in NGM plates seeded with *E. coli* OP50 using the Triglyceride Quantification Kit (Biovision; Mountain View, CA, USA). For the treatments, NGM plates were supplemented with 2 mg/mL of CTE or Orlistat (6 µg/mL) as a positive control. Worms at young adult stage were then collected and washed with M9 buffer. Supernatant was removed after worm setting, and 400 µL of triglyceride assay buffer was added to worm pellet. Worms were sonicated with a digital sonicator (Branson Ultrasonics Corp.; Danbury, CT, USA) at 10% of power for 30 s. To solubilize all TGs in the solution, samples were slowly heated twice at 90 °C for 5 min in a thermomixer (Thermo Fisher; Waltham, MA, USA). After centrifugation, 50 µL aliquots were used for the triglyceride assay following the manufacturer’s instructions. Protein content was measured using BCA Protein Assay Kit (Thermo Scientific; Rockford, IL, USA). Four different biological replicates were included for each condition in two independent experiments.

### 4.9. Murine Model of MetS

▪Animals

Forty 16-week-old C57/BL6J mice were housed two per cage and maintained in climate-controlled quarters under controlled conditions of humidity (50–60%) and temperature (22–24 °C), and with a 12 h light cycle. All of the experiments were conducted according to the European Union Legislation and with the approval of the Animal Care and Ethical Committee of the Community of Madrid (Madrid, Spain) (PROEX 214.1_20).

Mice were fed ad libitum and divided into three experimental groups: mice fed with a standard chow (Chow; *n* = 10), mice fed a high fat/high sucrose (HFHS) diet containing 58% kcal from fat with sucrose (HFHS; *n* = 10) and mice fed a high fat/high sucrose (HFHS) diet supplemented with 1.6% Complex Tea Extract (HFHS + CTE; *n* = 7). The customized diets were elaborated by the company Research Diets Inc. (New Brunswick, NJ, USA). The tea extract was added to the commercial high fat/high sucrose diet (HFHS) with reference D12331 at 1.6%. Both HFHS and HFHS + CTE were isocaloric. The diet with reference D11112201 was used as the standard diet (chow).

During treatment, body weight and solid and liquid intake were monitored once a week. After 20 weeks of treatment, all animals were sacrificed by an overdose of sodium pentobarbital (100 mg/kg) and killed by decapitation after overnight fasting. After euthanasia, the blood was collected in tubes with EDTA (1.5 mg/mL) and centrifuged at 3000 rpm for 20 min to obtain plasma. Liver, spleen, kidneys, adrenal glands and adipose tissue depots (epididymal, retroperitoneal, subcutaneous, brown and PVAT) were dissected, weighed and stored at −80 °C for later analysis.

▪Plasma measurements

Triglycerides, total cholesterol, low-density lipoprotein cholesterol (LDL-c) and high-density lipoprotein cholesterol (HDL-c) were measured in the plasma using commercial kits from Spin React S.A.U (Sant Esteve de Bas, Gerona, Spain) following the manufacturer’s instructions. Likewise, plasma concentrations of leptin, insulin and adiponectin were measured by ELISA kits (Merck Millipore, Dramstadt, Germany) following the manufacturer’s instructions. Sensitivity and intrassay variations were 0.05 ng/mL and 1.76–3.01% for leptin assay, 0.1 ng/mL and 1.92–7.64% for insulin assay and 0.2 ng/mL and 1.4–5.4% for adiponectin assay.

▪Homeostatic Model Assessment of Insulin Resistance (HOMA-IR)

Prior to sacrifice and after overnight fasting, glycemia was measured by venous tail puncture using GlucocardTM G (Arkray Factory, Inc., Koji Konan-cho, Koka, Japan). The HOMA-IR index was calculated through the following formula: fasting glucose (mg/dL) × (fasting insulin (ng/mL)/405).

▪Incubation of liver, gastrocnemius muscle and retroperitoneal adipose tissue explants in the presence/absence of Insulin (10^−6^ M)

In total, 100 mg explants were incubated with Dulbecco’s modified Eagle’s medium Gibco ((DMEM/F-12; 1:1 mix; Invitrogen, Carlsbad, CA, USA) plus 100 U/mL penicillin and 100 μg/mL streptomycin (Invitrogen, Carlsbad, CA, USA) in the presence/absence of insulin (10^−6^ M) (Sigma-Aldrich, St. Louis, MO, USA) at 37 °C in a 95% O_2_ and 5% CO_2_ incubator. After 15 min of incubation, all tissues were collected and stored at −80 °C for further analysis.

▪Protein quantification by Western Blot

In total, 100 mg of liver, gastrocnemius muscle and retroperitoneal adipose tissue was homogenized in 500 uL of RIPA buffer and centrifugated at 14,000 rpm for 20 min at 4 °C. The supernatant was collected, and the total protein content was measured by Bradford method (Sigma-Aldrich, St. Louis, MO, USA). For each assay, 10 µL of protein was loaded into each well of 10% acrylamide SDS gels (Bio-Rad, Hercules, CA, USA) and separated by electrophoresis. The proteins were then transferred to polyvinylidene difluoride (PVDF) membranes (Bio-Rad, Hercules, CA, USA). Transfer efficiency was determined by Ponceau red dyeing (Sigma-Aldrich, St. Louis, MO, USA). Membranes were then blocked either with tris-buffered saline (TBS) containing 5% (*w*/*v*) non-fat dried milk (for non-phosphorylated proteins) or with 5% BSA (for phosphorylated proteins) and incubated with the appropriate primary antibody for Akt (1:1000; # 04-796, Merk Millipore, Dramstad, Germany) or p-Akt (Ser 473) (1:500; #9271, Cell Signaling Technology, Danvers, MA, USA) at 4 °C overnight. Membranes were subsequently washed and incubated with the secondary antibody conjugated with peroxidase (1:2000; Pierce, Rockford, IL, USA). Peroxidase activity was visualized by chemiluminescence and quantified by densitometry using BioRad Molecular Imager ChemiDoc XRS System (Hercules, CA, USA). For each sample, relative protein expression levels were calculated in relation to protein expression levels in mice fed with Chow.

▪Gene expression analysis by qPCR

The total RNA was extracted from 100 mg of liver, gastrocnemius muscle and retroperitoneal adipose tissue using the Tri-Reagent protocol [79] and quantified with Nanodrop 2000 (Thermo Fisher Scientific, Hampton, NH, USA). cDNA was then synthesized from 1 µg of total RNA using a high-capacity cDNA reverse transcription kit (Applied Biosystems, Foster City, CA, USA).

Assay-on-demand kits (Applied Biosystems, Foster City, CA, USA) were used for quantitative real-time polymerase chain reaction (qPCR). TaqMan Universal PCR Master Mix (Applied Biosystems, Foster City, CA, USA) was used for amplification according to the manufacturer’s instructions in a Step One System (Applied Biosystems, Foster City, CA, USA).

In liver, gastrocnemius muscle and retroperitoneal adipose tissue, the gene expression of insulin receptor (*Ins-r*) (Mm01211875_m1), glycogen sinthase 1 (*Gys1*) (Mm01962575_s1), monocyte chemoattractant protein (*Mcp-1*) (Mm00441242_m1), interleukin-1 beta (*IL-1β*) (Mm00434228_m1), interleukin-6 (*IL-6*) (Mm00446190_m1), interleukin 10 (*IL-10*) (Mm01288386_m1), tumor necrosis factor-alpha (*Tnf-α*) (Mm00443258_m1), NADPH oxidase-1 (*Nox-1*) (Mm00549170_m1), NADPH oxidase-4 (*Nox-4*) (Mm00479246_m1), superoxide dismutase 1 (*Sod-1*) (Mm01344233_g1), Glutathione Peroxidase 3 (*Gpx-3*) (Mm00492427_m1), glutathione reductase (*Gsr*) (Mm00439154_m1), heme oxygenase 1 (*Ho-1*) (Mm00516005_m1), aryl-hydrocarbon receptor (*Ahr*) (Mm00478932_m1), aryl hydrocarbon receptor nuclear translocator (*Arnt*) (Mm00507836_m1) and Nuclear factor erythroid 2-related factor 2 (*Nrf2*) (Mm00477784_m1) were measured. In addition, the following genes were also analyzed in retroperitoneal adipose: fatty acid synthetase (*Fasn*) (Mm00662319_m1), lipoprotein lipase (*Lpl*) (Mm00434764_m1), hormone-sensitive lipase (*Hsl*) (Mm00495359_m1), peroxisome proliferator-activator receptor-γ (*Ppar-γ*) (Mm00440940_m1), beta-3 adrenergic receptor (*β3-Adr*) (Mm02601819_g1), leptin receptor (*Ob-r*) (Mm00440181_m1), peroxisome proliferator-activated receptor gamma coactivator 1-alpha (*Pgc-1α*) (Mm01208835_m1) and uncoupling protein 1 (*Ucp-1*) (Mm01244861_m1).

The values were normalized to the housekeeping gene Hypoxanthine Phosphoribosyltransferase 1 (*Hprt-1*) (Mm03024075_m1). In order to determine the relative expression levels, the ΔΔCT method was used [80]. All the data are expressed as percentages vs. the control group (Chow).

▪Adipocyte Size

Retroperitoneal adipose tissue samples were fixed overnight in 4% paraformaldehyde and embedded in paraffin wax. Then, 5 µm sections were cut with a microtome, mounted into slides and stained with Harris Hematoxylin and Eosin. Images were acquired using a Leica light microscope with a 10× objective (Wetzlar, Germany). To determine the adipocyte size, the area of each adipocyte was measured using FIJI for Windows 36 bit (NIH, Bethesda, MA, USA).

▪Statistical Analysis

One-way ANOVA followed by a Bonferroni post hoc test was used for the statistical data analysis using GraphPad Prism 8.0 (San Diego, CA, USA). All the values are expressed as means ± the standard error of the mean (SEM). A *p*-value of ≤0.05 was considered statistically significant.

## Figures and Tables

**Figure 1 ijms-24-08521-f001:**
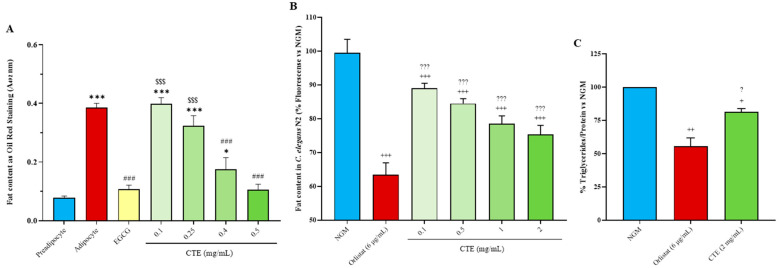
Fat content measured by Oil Red O staining in preadipocytes 3T3-L1 cells and differentiated 3T3-L1 cells treated or not with EGCG and CTE (**A**), fat content as % of fluorescence produced by Nile red staining vs. NGM (**B**), and triglyceride content as % vs. NGM (**C**) in nematodes fed with *Escherichia coli* OP50 and orlistat or CTE. Data are represented as the mean ± SD. * *p* < 0.05 vs. preadipocytes; *** *p* < 0.001 vs. preadipocytes; ### *p* < 0.001 vs. non-treated adipocytes; $$$ *p* < 0.001 vs. EGCG; + *p* < 0.05 vs. NGM; ++ *p* < 0.01 vs. NGM; +++ *p* < 0.001 vs. NGM; ? *p* < 0.05 vs. Orlistat; ??? *p* < 0.001 vs. Orlistat. CTE: Complex Tea Extract; EGCG: epigallocatechin gallate; NGM: nematode growth media.

**Figure 2 ijms-24-08521-f002:**
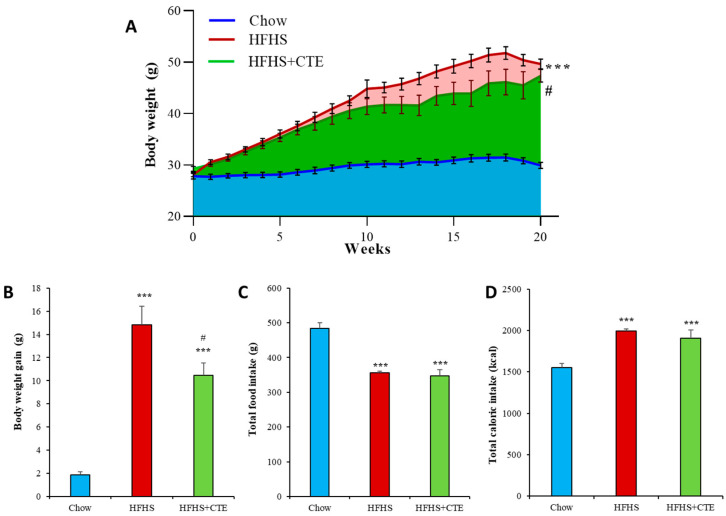
Evolution of body weight during the 20 weeks of treatment (**A**). Body weight gain during the treatment (**B**). Total food intake per mouse during the treatment (**C**). Total caloric intake per mouse during the treatment (**D**) of mice fed a standard chow (Chow), a high-fat diet/sucrose diet (HFHS) and high-fat diet/sucrose diet supplemented with Complex Tea Extract (HFHS + TCE). Values are represented as mean ± SEM; *n* = 8–10 mice/group. *** *p* < 0.001 vs. chow; # *p* < 0.05 vs. HFHS.

**Figure 3 ijms-24-08521-f003:**
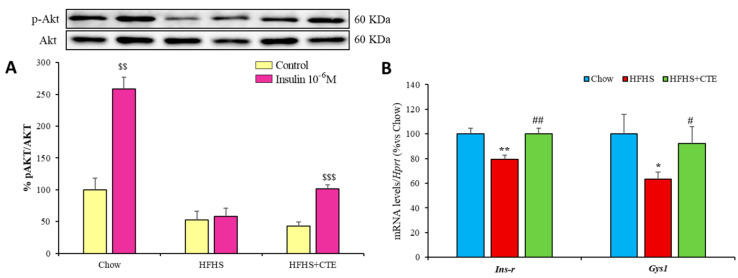
p-Akt/Akt ratio (**A**) in liver explants of mice fed a standard chow (Chow), a high-fat diet/sucrose diet (HFHS) and high-fat diet/sucrose diet supplemented with Complex Tea Extract (HFHS + TCE) after 15 min of incubation with or without 10^−6^ M insulin. mRNA levels of insulin receptor and Glycogen synthase 1 (**B**) in liver. Values are represented as mean ± SEM; *n* = 8–10 mice/group. $$ *p* < 0.01 vs. Control; $$$ *p* < 0.001 vs. Control; * *p* < 0.05 vs. chow; ** *p* < 0.01 vs. chow; # *p* < 0.05 vs. HFHS; ## *p* < 0.01 vs. HFHS.

**Figure 4 ijms-24-08521-f004:**
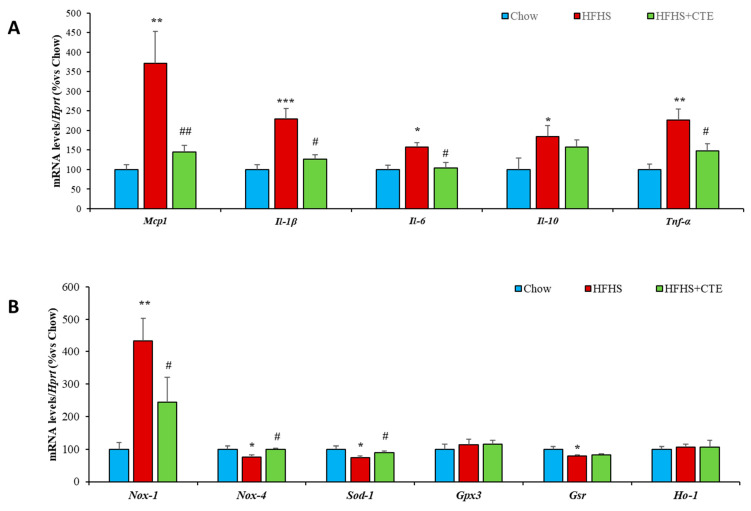
mRNA levels of Monocyte Chemotactic Protein-1, Interleukin 1β, 6, 10 and Tumor Necrosis Factor α (**A**), NADPH oxidase 1 and 4, Super Oxide Dismutase 1, Glutathione Peroxidase and Reductase, Hemoxigenase-1 (**B**) in hepatic tissue of mice fed a standard chow (Chow), a high-fat diet/sucrose diet (HFHS) and high-fat diet/sucrose diet supplemented with Complex Tea Extract (HFHS + TCE). Values are represented as mean ± SEM; *n* = 8–10 mice/group. * *p* < 0.05 vs. chow; ** *p* < 0.01 vs. chow; *** *p* < 0.001 vs. chow; # *p* < 0.05 vs. HFHS; ## *p* < 0.01 vs. HFHS.

**Figure 5 ijms-24-08521-f005:**
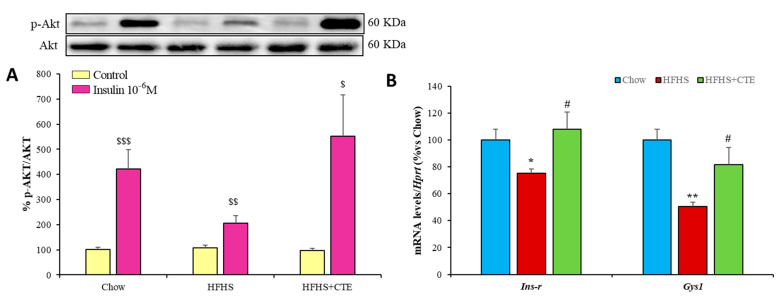
pAkt/Akt ratio (**A**) in gastrocnemius muscle after 15 min of explant incubation with or without 10^−6^ M insulin of mice fed a standard chow (Chow), a high-fat diet/sucrose diet (HFHS) and high-fat diet/sucrose diet supplemented with Complex Tea Extract (HFHS + TCE). mRNA levels of insulin receptor and Glycogen synthase 1 (**B**) in gastrocnemius muscle. Values are represented as mean ± SEM; *n* = 8–10 mice/group. $ *p* < 0.05 vs. Control; $$ *p* < 0.01 vs. Control; $$$ *p* < 0.001 vs. Control; * *p* < 0.05 vs. chow; ** *p* < 0.01 vs. chow; # *p* < 0.01 vs. HFHS.

**Figure 6 ijms-24-08521-f006:**
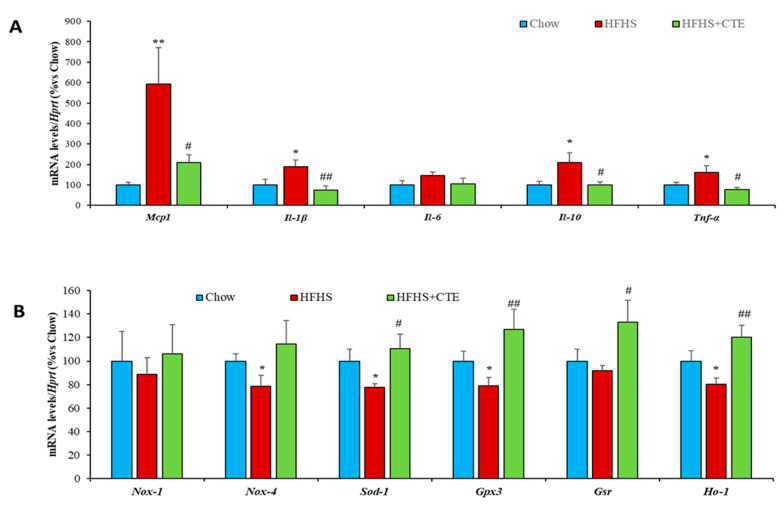
mRNA levels of Monocyte Chemotactic Protein-1, Interleukin 1β, 6, 10 and Tumor Necrosis Factor α (**A**), NADPH oxidase 1 and 4, Super Oxide Dismutase 1, Glutathione Peroxidase and Reductase, Hemoxygenase-1 (**B**) in gastrocnemius muscle of mice fed a standard chow (Chow), a high-fat diet/sucrose diet (HFHS) and high-fat diet/sucrose diet supplemented with Complex Tea Extract (HFHS + TCE). Values are represented as mean ± SEM; *n* = 8–10 mice/group. * *p* < 0.05 vs. chow; ** *p* < 0.01 vs. chow; # *p* < 0.05 vs. HFHS; ## *p* < 0.01 vs. HFHS.

**Figure 7 ijms-24-08521-f007:**
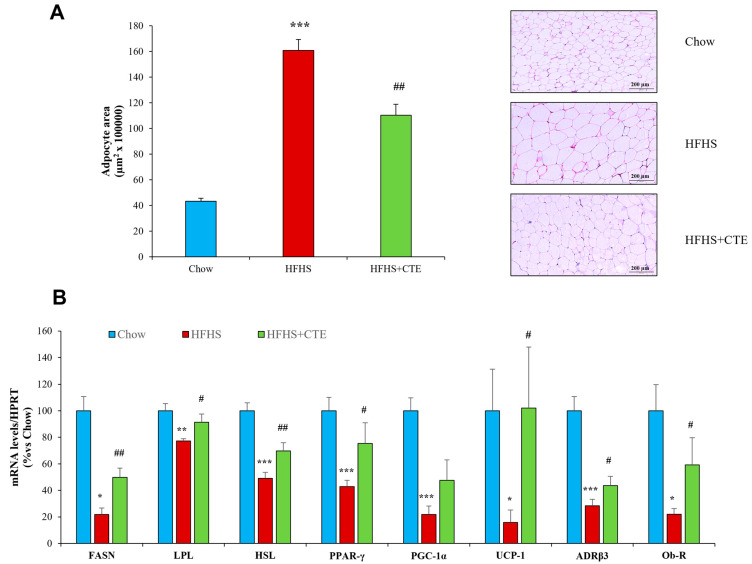
Size of adipocytes and representative images of H/E dying in sections of visceral adipose tissue (**A**). The scale bar is equivalent to 200 µm. mRNA levels of Fatty acid synthase, Lipoprotein lipase, Hormone-sensitive lipase, peroxisome proliferator activated receptor, PPAR-γ coactivator-1α, Uncoupling Protein-1, β-3-adrenergic receptor and Leptin receptor (**B**) in retroperitoneal adipose tissue of mice fed a standard chow (Chow), a high-fat diet/sucrose diet (HFHS) and high-fat diet/sucrose diet supplemented with Complex Tea Extract (HFHS + TCE). Values are represented as mean ± SEM; *n* = 8–10 mice/group. * *p* < 0.05 vs. chow; ** *p* < 0.01 vs. chow; *** *p* < 0.001 vs. chow; # *p* < 0.05 vs. HFHS; ## *p* < 0.01 vs. HFHS.

**Figure 8 ijms-24-08521-f008:**
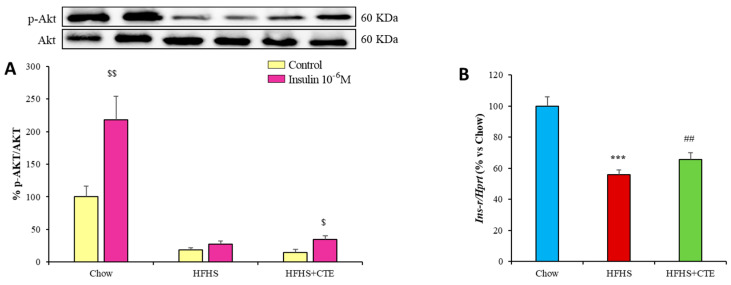
pAkt/Akt ratio (**A**) in retroperitoneal adipose tissue of mice fed a standard chow (Chow), a high-fat diet/sucrose diet (HFHS) and high-fat diet/sucrose diet supplemented with Complex Tea Extract (HFHS + TCE) after 15 min of explant incubation with or without 10^−6^ M insulin. mRNA levels of insulin receptor in retroperitoneal adipose tissue (**B**). The scale bar is equivalent to 200 µm. Values are represented as mean ± SEM; *n* = 8–10 mice/group. $ *p* < 0.05 vs. Control; $$ *p* < 0.01 vs. Control; *** *p* < 0.001 vs. chow; ## *p* < 0.01 vs. HFHS.

**Figure 9 ijms-24-08521-f009:**
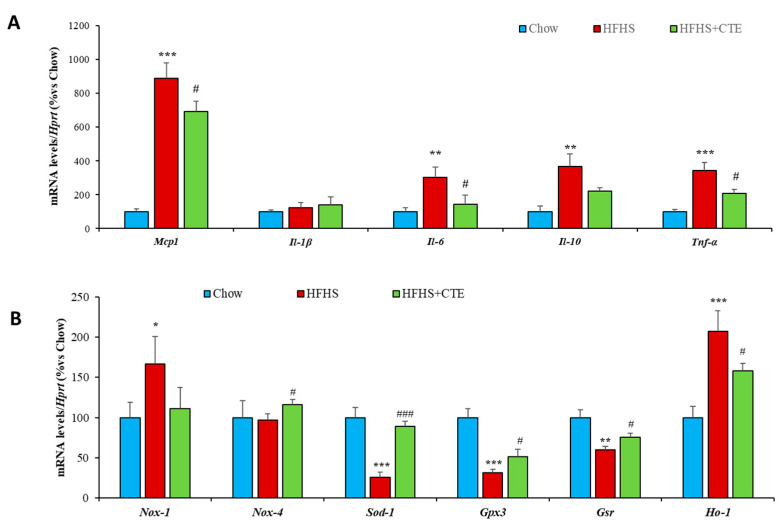
mRNA levels of Monocyte Chemotactic Protein-1, Interleukin 1β, 6, 10 and Tumor Necrosis Factor α (**A**), NADPH oxidase 1 and 4, Super Oxide Dismutase 1, Glutathione Peroxidase and Reductase, and Hemoxigenase-1 (**B**) in retroperitoneal adipose tissue of mice fed a standard chow (Chow), a high-fat diet/sucrose diet (HFHS) and high-fat diet/sucrose diet supplemented with Complex Tea Extract (HFHS + TCE)**.** Values are represented as mean ± SEM; *n* = 8–10 mice/group. ** p* < 0.05 vs. chow; *** p* < 0.01 vs. chow; **** p* < 0.001 vs. chow; *# p* < 0.05 vs. HFHS; *### p* < 0.001 vs. HFHS.

**Figure 10 ijms-24-08521-f010:**
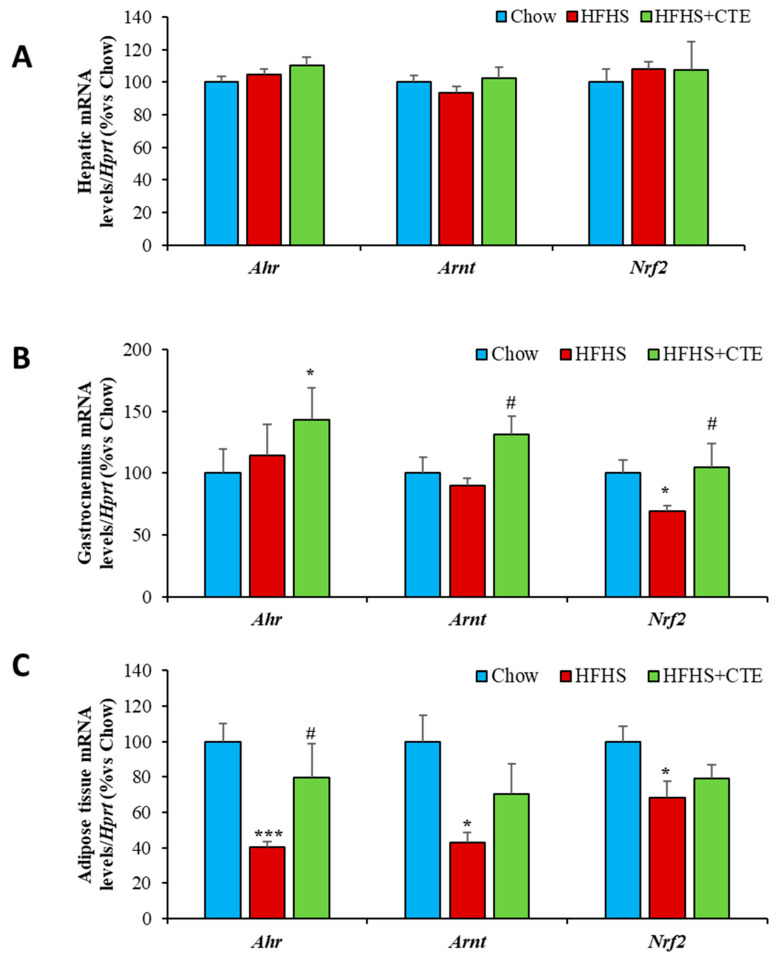
Hepatic (**A**), gastrocnemius muscle (**B**) and retroperitoneal adipose tissue (**C**) mRNA levels of Aryl Hydrocarbon Receptor, Aryl Hydrocarbon Receptor Nuclear and Nuclear factor erythroid 2-related factor 2 (**C**) of mice fed a standard chow (Chow), a high-fat diet/sucrose diet (HFHS) and high-fat diet/sucrose diet supplemented with Complex Tea Extract (HFHS + TCE)**.** Values are represented as mean ± SEM; *n* = 8–10 mice/group. ** p* < 0.05 vs. chow; **** p* < 0.001 vs. chow; *# p* < 0.05 vs. HFHS.

**Table 1 ijms-24-08521-t001:** Bioactive composition (%, dry basis) of ADM^®^ Complex Tea Extract analyzed by HPLC-PAD.

Bioactive Composition	Tea Complex
Gallic acid	0.77 ± 0.03
Theobromine	0.26 ± 0.01
Gallocatechin	0.43 ± 0.04
Theophyllyne	0.01 ± 0.00
Epigallocatechin	0.74 ± 0.04
Catechin	0.00 ± 0.00
Caffeine	6.53 ± 0.13
Epicatechin	0.63 ± 0.03
EGCg	5.98 ± 0.26
Gallocatechin-3-gallate	1.76 ± 0.07
Epicatechin-3-gallate	2.49 ± 0.09
Catechin-3-gallate	0.63 ± 0.04
Theaflavin	0.06 ± 0.00
Theaflavin-3-monogallate	0.03 ± 0.00
Theaflavin-3′-monogallate	0.01 ± 0.00
Theaflavin-3,3′-gallate	0.03 ± 0.00
Monomeric Flavan-3-ols	12.66 ± 0.45
Methyl xanthines	6.80 ± 0.13
Oligomeric Flavan-3-ols (Theaflavins)	0.13 ± 0.00
Total bioactive components (flavan-3-ols, methylxanthines and gallic acid)	20.36 ± 0.44

**Table 2 ijms-24-08521-t002:** Organ weights.

Weight (mg/cm)	Chow	HFHS	HSHS + CTE
Liver	535.0 ± 11.3	888.5 ± 72.9 ***	758.6 ± 57.1 ***
Spleen	34.9 ± 2.3	48.4 ± 2.4 **	41.3 ± 2.4 *^#^
Kidneys	171.2 ± 5.1	202.5 ± 3.7 ***	207.9 ± 6.7 ***
Adrenal glands	1.5 ± 0.1	2.1 ± 0.2 *	1.9 ± 0.2 *
Epidydimal visceral adipose tissue	357.8 ± 40.0	1577.0 ± 87.2 ***	1763.3 ± 42.0 ***
Retroperitoneal visceral adipose tissue	108.8 ± 21.5	713.6 ± 32.5 ***	578.8 ± 17.5 ***^###^
Lumbar subcutaneous adipose tissue	143.5 ± 17.9	1463.2 ± ***	1185.8 ± 108.5 ***^#^
Interscapular brown adipose tissue	40.8 ± 3.2	142.1 ± 11.1 ***	87.5 ± 9.6 ***^###^
Periaortic adipose tissue	3.5 ± 0.4	11.6 ± 1.7 ***	8.2 ± 0.9 ***
Soleus	5.0 ± 0.3	5.91 ± 0.3 *	5.82 ± 0.3 ***
Gastrocnemius	70.0 ± 1.4	73.6 ± 1.9	74.29 ± 1.9 ***

Weights of liver, spleen, kidneys, adrenal glands, epididymal visceral adipose tissue, retroperitoneal visceral adipose tissue, lumbar subcutaneous adipose tissue, interscapular brown adipose tissue, periaortic adipose tissue, soleus and gastrocnemius of mice fed a standard chow (Chow), a high-fat diet/sucrose diet (HFHS) and high-fat diet/sucrose diet supplemented with Complex Tea Extract (HFHS + TCE). Data are represented as mean value ± SEM; *n* = 8–10 mice/group. * *p* < 0.05 vs. Chow; ** *p* < 0.01 vs. Chow; *** *p* < 0.001 vs. Chow; # *p* < 0.05 vs. HFHS; ### *p* < 0.001 vs. HFHS.

**Table 3 ijms-24-08521-t003:** Glycaemia, insulin, adiponectin and leptin plasma levels, and Homeostatic Model Assessment of Insulin Resistance (HOMA-IR).

	Chow	HFHS	HSHS + CTE
Glycaemia (mg/dl)	102.9 ± 4.9	133.8 ± 6.1 ***	145.4 ± 5.4 ***
Insulin (ng/mL)	2.5 ± 0.8	8.5 ± 1.2 ***	4.6 ± 0.8 *^#^
Adiponectin (ng/mL)	9683.6 ± 374.1	7076.4 ± 613.7 ***	8299 ± 324 **^#^
Leptin (ng/mL)	2.2 ± 0.6	44.0 ± 3 ***	30.2 ± 1.3 ***^##^
HOMA-IR	0.7 ± 0.2	2.9 ± 0.5 ***	1.6 ± 0.3 ***^#^

Data from mice fed a standard chow (Chow), a high-fat diet/sucrose diet (HFHS) and high-fat diet/sucrose diet supplemented with Complex Tea Extract (HFHS + TCE). Data are represented as mean value ± SEM; *n* = 8–10 mice/group. * *p* < 0.05 vs. Chow; ** *p* < 0.01 vs. Chow; *** *p* < 0.001 vs. Chow; # *p* < 0.05 vs. HFHS; ## *p* < 0.01 vs. HFHS.

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
