# Peer review of "Supplementation with a New Standardized Extract of Green and Black Tea Exerts Antiadipogenic Effects and Prevents Insulin Resistance in Mice with Metabolic Syndrome"

_ijms, 2023, doi:10.3390/ijms24108521_

Round 1

Reviewer 1 Report

In this study, authors  examined the effect of tea extract on obesity, which had been variable, and propose a standard product. The experiment is carefully done and the design is appropriate. The results are valuable. On the other hand, the manuscript had some questions.

1) There is a problem with the format of the manuscript. This manuscript is very difficult to read.

1-1) line 198: Isn't "Results" a mistake? The following text looks like "Materials and Methods". Additionally, line 395 has "Results" again.

1-2) Table 2: The title is too long. The title should be simpler. Write any necessary explanations in the footnote.

1-3) Figures: All figures should have a concise title.

1-4) Please insert the figure in the appropriate position.

2) The authors propose a mixture of green and black tea extracts as standard of tea extract. Why did authors employ mixture? Green tea and black tea have different ingredients. And they also have different physiological activities. In particular, catechins are different between black tea and green tea. Doesn't it complicate the discussion?

3) Did data shown in table 1 obteined from manufacturere? Or the data was measured by authors ? If the data were measured by the authors, they should be shown in "Results".

4) According to table 1, CTE contains substantial amounts of caffeine. Were there any effects of caffeine on animals and cells? Caffeine is one of the main ingredients in tea. The effects of caffeine should also be discussed.

Author Response

Reviewer 1

In this study, authors  examined the effect of tea extract on obesity, which had been variable, and propose a standard product. The experiment is carefully done and the design is appropriate. The results are valuable. On the other hand, the manuscript had some questions.

1) There is a problem with the format of the manuscript. This manuscript is very difficult to read.

1-1) line 198: Isn't "Results" a mistake? The following text looks like "Materials and Methods". Additionally, line 395 has "Results" again.

We thank the reviewer for this appreciation. It was a mistake that has already been corrected.

1-2) Table 2: The title is too long. The title should be simpler. Write any necessary explanations in the footnote.

It has been corrected as suggested by the reviewer.

1-3) Figures: All figures should have a concise title.

All the titles have been moved to the footnote.

1-4) Please insert the figure in the appropriate position.

All the figures are now inserted in the correct position

2) The authors propose a mixture of green and black tea extracts as standard of tea extract. Why did authors employ mixture? Green tea and black tea have different ingredients. And they also have different physiological activities. In particular, catechins are different between black tea and green tea. Doesn't it complicate the discussion?

We really appreciate the questions raised by the reviewer in relation to the green and black tea blend, named in the manuscript as Complex Tea Extract (CTE). First of all, we would like to clarify that CTE is not a mixture but a proprietary combination of green and black tea leaves which are co-extracted in a defined proportion (according to the initial bioactive composition) and processed to a finished powder.

As it is widely known that green and black teas are different not only from an analytical and sensory point of view, but also for their biological effects. For this reason, we think that the combination of the bioactive compounds of both types of teas in one extract is one of the most interesting and novel aspects of this new product since the separate effects of both types of tea have already been widely studied. Indeed, there are several studies in the literature focusing just on the functional properties green tea (more concentrated in monomeric flavan-3-ols) and   black tea (concentrated into oligomeric (theaflavins) and polymeric flavan-3-ols (thearubigins) separately. However, to our knowledge, this is the first manuscript that uses an industrial tea extract standardized into the main bioactive components from both green and black tea extracts, including bioactive and volatile compounds.

A paragraph regarding this issue has been added to the results section (Lines 403-405)

3) Did data shown in table 1 obteined from manufacturer? Or the data was measured by authors ? If the data were measured by the authors, they should be shown in "Results".

The results were obtained by the authors not by the manufacturer. In fact, data from the manufacturer (quality control) do not consider individual composition, just Total Flavan-3-ols and Total Xantines. To better clarify this issue, a paragraph in lines 216-219 has been added (Lines 388-391).

4) According to table 1, CTE contains substantial amounts of caffeine. Were there any effects of caffeine on animals and cells? Caffeine is one of the main ingredients in tea. The effects of caffeine should also be discussed.

Caffeine effects were investigated in previous work (Martinez-Saez et al., 2014) in C. elegans fat reduction assay. In these experiments, a dose-response effect on reducing accumulation of body fat was observed for caffeine reaching a 30 % fat reduction at 25 µM (4,86 µg/mL). In agreement the with C elegans experiments, we have data from our lab that show that caffeine was also able to inhibit lipid accumulation in 3T3-L1 cells during the differentiation process displaying maximum inhibition at 2 mM (0,39 mg/mL) as it is shown in the graph below

Thus, taking into account the antiadipogenic effects of caffeine, the presence of this bioactive compound in the CTE may contribute, at least in part, to the beneficial effects of the extract reducing adiposity.

A paragraph stating the possible effects of caffeine of adiposity is now present in the Discussion section (Lines 682-685).

Reviewer 2 Report

The manuscript entitled “Supplementation with a new standardized extract of green and 2 black tea exerts antiadipogenic effects and prevents insulin resistance in mice with metabolic syndrome” refers to actual and interesting issue.
Article deals with very important problem of the 21st century, i.e. the insulin resistance.
The described tests involved with application of a standardized extract of green and black tea in daily diet shown the preventive activity against the development of insulin resistance in mice.
The resulted effects are very promising for humans.
The obtained results are interesting taking into account the practical significance of the considered issue. Nevertheless, I have some doubt refers to the applied statistical analysis.
I think that the manuscript should be published after minor correction and explanation.
1. Was the normality of the distribution of the variable checked  for example, the Shapiro-Wilk test?
2. Did the authors use the Student's Test to compare the datasets?

Author Response

Reviewer 2

The manuscript entitled “Supplementation with a new standardized extract of green and 2 black tea exerts antiadipogenic effects and prevents insulin resistance in mice with metabolic syndrome” refers to actual and interesting issue.

Article deals with very important problem of the 21st century, i.e. the insulin resistance.

The described tests involved with application of a standardized extract of green and black tea in daily diet shown the preventive activity against the development of insulin resistance in mice.

The resulted effects are very promising for humans.

The obtained results are interesting taking into account the practical significance of the considered issue.

We thank the reviewer for his/her positive comments about the manuscript.

Nevertheless, I have some doubt refers to the applied statistical analysis. I think that the manuscript should be published after minor correction and explanation.

  1. Was the normality of the distribution of the variable checked for example, the Shapiro-Wilk test?

Yes, the statistical software used calculates first if the data have a normal distribution.

  1. Did the authors use the Student's Test to compare the datasets?

No, as it is stated in the material and methods section, post-hoc comparisons were made using the Bonferroni test.

Reviewer 3 Report

I have no comments on the work as it is written and organized very well, even if conflicts of interest are declared, they are still very limited. Therefore, after an accurate evaluation, I consider the publication suitable.

Author Response

We thank the reviewer for his/her positive comments about the manuscript.

Round 2

Reviewer 1 Report

I am satisfied with the author's response and revision.